# Regulation of TRIF-mediated innate immune response by K27-linked polyubiquitination and deubiquitination

Xin Wu[1,2], Caoqi Lei[2], Tian Xia [1,2], Xuan Zhong[1,2], Qing Yang[1] & Hong-Bing Shu [1,2]

TIR domain-containing adaptor inducing interferon-β (TRIF) is an essential adaptor protein required for innate immune responses mediated by Toll-like receptor (TLR) 3- and TLR4. Here we identify USP19 as a negative regulator of TLR3/4-mediated signaling. USP19 deficiency increases the production of type I interferons (IFN) and proinflammatory cytokines induced by poly(I:C) or LPS in vitro and in vivo. $Usp19^{-/-}$ mice have more serious inflammation after poly(I:C) or LPS treatment, and are more susceptible to inflammatory damages and death following *Salmonella typhimurium* infection. Mechanistically, USP19 interacts with TRIF and catalyzes the removal of TRIF K27-linked polyubiquitin moieties, thereby impairing the recruitment of TRIF to TLR3/4. In addition, the RING E3 ubiquitin ligase complex Cullin-3-Rbx1-KCTD10 catalyzes K27-linked polyubiquitination of TRIF at K523, and deficiency of this complex inhibits TLR3/4-mediated innate immune signaling. Our findings thus reveal TRIF K27-linked polyubiquitination and deubiquitination as a critical regulatory mechanism of TLR3/4-mediated innate immune responses.

---

[1] Department of Infectious Diseases, Medical Research Institute, Zhongnan Hospital of Wuhan University, Wuhan University, 430071 Wuhan, China.
[2] Department of Cell Biology, College of Life Sciences, Wuhan University, 430072 Wuhan, China. Correspondence and requests for materials should be addressed to H.-B.S. (email: shuh@whu.edu.cn)

Toll-like receptors (TLR) are evolutionarily conserved receptors which play important roles in host defense against a wide variety of pathogens. Recognition of conserved microbial structural motifs known as pathogen-associated molecular patterns (PAMPs) by TLRs initiates signaling cascades that lead to activation of transcription factors such as interferon regulatory factor 3 (IRF3) and nuclear factor κB (NF-κB), which culminate in transcription of downstream effector genes and induction of innate immune and inflammatory responses[1,2].

TLRs are type I transmembrane proteins and distinct in their cellular localization and ligand recognition. TLRs contain an N-terminal leucine-rich repeats domain, which is responsible for ligand recognition, a transmembrane domain and an intracellular Toll/interleukin 1 (IL-1) receptor (TIR) domain for recruitment of downstream TIR domain-containing adapters, such as MyD88 and TIR domain-containing adapter inducing IFN-β (TRIF, also called TICAM-1)[1,2]. Most TLRs signal through MyD88, but TLR3 signals through TRIF, while TLR4 utilizes both MyD88 and TRIF for signaling, giving rise to specificity in TLR3/4-mediated signaling. TLR3 localizes both on the plasma membrane and intracellular endosomes, and is essential for recognition of viral dsRNA as well as its synthetic analog poly(I:C)[3]. TRIF-deficient mice are defective in TLR3-mediated induction of type I IFNs and inflammatory cytokines, suggesting that TRIF is required for TLR3-mediated signaling[4,5]. TLR4, which recognizes lipopolysaccharide (LPS) of Gram-negative bacteria, is the only receptor which signals through MyD88-dependent pathway to activate NF-κB and TRIF-dependent pathway to activate both NF-κB and IRF3. Consistently, LPS-induced IRF3 but not NF-κB activation is impaired in TRIF-deficient cells, whereas LPS-induced NF-κB but not IRF3 is inhibited in MyD88-deficient cells[4,6–8].

TRIF is a 712 aa protein which contains an N-terminal proline-rich region (PRR), a middle TIR domain and a C-terminal RIP homotypic interaction motif (RHIM)[4,5]. The PRR of TRIF associates with TRAF3 and TBK1, which is required for TRIF-mediated IRF3 activation[2,9]. TRIF also activates NF-κB through its RHIM, which mediates the recruitment of RIP1 and TRAF6. In addition, TRIF is capable of inducing apoptosis through RIP1, FADD, and caspase-8[4,10–12]. Previously, it has been shown that TRIF is heavily modified and regulated by polyubiquitination. Two E3 ubiquitin ligases, WWP2 and TRIM38, have been shown to mediate K48-linked polyubiquitination of TRIF and inhibit TLR3/4-mediated innate immune responses[13,14]. TRIM8 mediates K6- and K33-linked polyubiquitination of TRIF, which impairs its recruitment of TBK1, therefore, negatively regulates TLR3/4-mediated innate immune responses[15]. Whether TRIF is regulated by other types of polyubiquitination is unknown.

In this study, we identify ubiquitin-specific peptidase 19 (USP19) as a negative regulator of TLR3/4-mediated signaling in expression screens. Gene knockout studies demonstrate that USP19 negatively regulates TLR3/4-mediated innate immune and inflammatory responses in cells and mice. Biochemical studies indicate that TRIF is modified by K27-linked polyubiquitination, which is critical for the recruitment of TRIF to TLR3/4 following ligand stimulation, and USP19 specifically removes K27-linked polyubiquitin moieties of TRIF after TLR3/4 activation. Our results also suggest that the RING E3 ubiquitin ligase complex Cullin-3–Rbx1–KCTD10 is responsible for catalyzing K27-linked polyubiquitination of TRIF at K523, and its deficiency inhibits TLR3/4-mediated innate immune signaling. These findings suggest that K27-linked polyubiquitination and deubiquitination of TRIF is a critical regulatory mechanism for TLR3/4-mediated signaling, and the Cullin-3-Rbx1-KCTD10 and USP19 ubiquitin ligase/deubiquitinating enzyme pair has an important function in fine-tuning TRIF-mediated innate immune and inflammatory responses.

## Results

**USP19 negatively regulates TLR3/4-mediated signaling.** In an attempt to identify deubiquitinating enzymes (DUBs) that regulate TLR3-mediated signaling, we screened 25 DUBs by IFN-β promoter reporter assays, and identified USP19 as a candidate. Overexpression of USP19 inhibited poly(I:C)-induced activation of the IFN-β promoter, ISRE, and NF-κB in a dose-dependent manner in 293-TLR3 cells (Supplementary Fig. 1a). In similar assays, USP19 also inhibited LPS-induced signaling in 293-TLR4 cells (Supplementary Fig. 1b). Overexpression of USP19 did not affect IFN-γ-triggered activation of the IRF1 promoter in HEK293 cells (Supplementary Fig. 1c). Quantitative PCR (qPCR) analysis indicated that overexpression of USP19 inhibited poly(I:C)-induced transcription of downstream antiviral genes including *IFNB1, ISG56, TNF*, and *CXCL10* (Supplementary Fig. 1d) and phosphorylation of TBK1, p65, and IRF3 (Supplementary Fig. 1e) in 293-TLR3 cells. These results suggest that overexpression of USP19 inhibits poly(I:C)- and LPS-triggered IRF3 and NF-κB activation.

We next examined the role of endogenous USP19 in regulation of TLR3/4-mediated signaling. We generated USP19-deficient (USP19-KO) 293-TLR3, 293-TLR4, and RAW264.7 cells by CRISPER/Cas9 technology[16]. As shown in Fig. 1, the USP19 protein was detected as two bands by immunoblots but not detectable in the respective KO cells. The simplest explanation is that this antibody can detect two different splice variants of USP19, or alternatively, the lower band represents a cleaved fragment of USP19 isoform 1. qPCR experiments showed that USP19-deficiency in 293-TLR3 cells potentiated poly(I:C)-induced transcription of *IFNB1, ISG56, TNF* and *CXCL10* genes (Fig. 1a). Similarly, USP19-deficiency also potentiated LPS-induced transcription of downstream genes in 293-TLR4 (Fig. 1b) and RAW264.7 cells (Fig. 1c). USP19-deficiency had no marked effects on Sendai virus (SeV, a RNA virus)-, herpes simplex virus 1 (HSV-1, a DNA virus)-, peptidoglycan (PGN, a synthetic TLR2 Ligand)-, or R848 (a synthetic TLR7/8 Ligand)-induced transcription of *Ifnb1, Tnf*, and *Cxcl10* genes in RAW264.7 cells (Supplementary Fig. 2). These data suggest that endogenous USP19 specifically inhibits TLR3/4-mediated signaling in human and mouse cell lines.

To investigate the functions of USP19 in vivo, we generated USP19-deficient mice by CRISPR/Cas9 technology. We firstly prepared bone marrow-derived macrophages (BMDMs), dendritic cells (DCs) and mouse lung fibroblasts (MLFs) from wild-type and USP19-deficient mice. Consistently with above experiments, we found that USP19-deficiency potentiated poly(I:C)- and LPS-induced transcription of downstream effector genes in BMDMs (Fig. 2a) and DCs (Supplementary Fig. 3a), but had no marked effects on SeV-, HSV-1-, PGN-, or R848-induced transcription of downstream effector genes in BMDMs (Supplementary Fig. 3b). Consistently, poly(I:C)- and LPS-induced phosphorylation of TBK1, IRF3, p65, and IκBα, which are hall-marks of IRF3 and NF-κB activation pathways, were markedly increased in *Usp19*$^{-/-}$ BMDMs compared with wild-type cells (Fig. 2b). In contrast, R848-triggered phosphorylation of p65 and IκBα were similar in *Usp19*$^{-/-}$ and *Usp19*$^{+/+}$ BMDMs (Supplementary Fig. 3c). Collectively, these results suggest that USP19 negatively regulates TLR3/4-mediated induction of downstream effector genes in primary mouse immune cells or fibroblasts.

**USP19-deficiency enhances TLR3/4-mediated response in vivo.** We next investigated the function of USP19 in the regulation of TLR3/4-mediated innate immune and inflammatory responses in vivo. Age- and sex-matched *Usp19*$^{+/+}$ and *Usp19*$^{-/-}$ mice were intraperitoneally injected with poly(I:C) plus

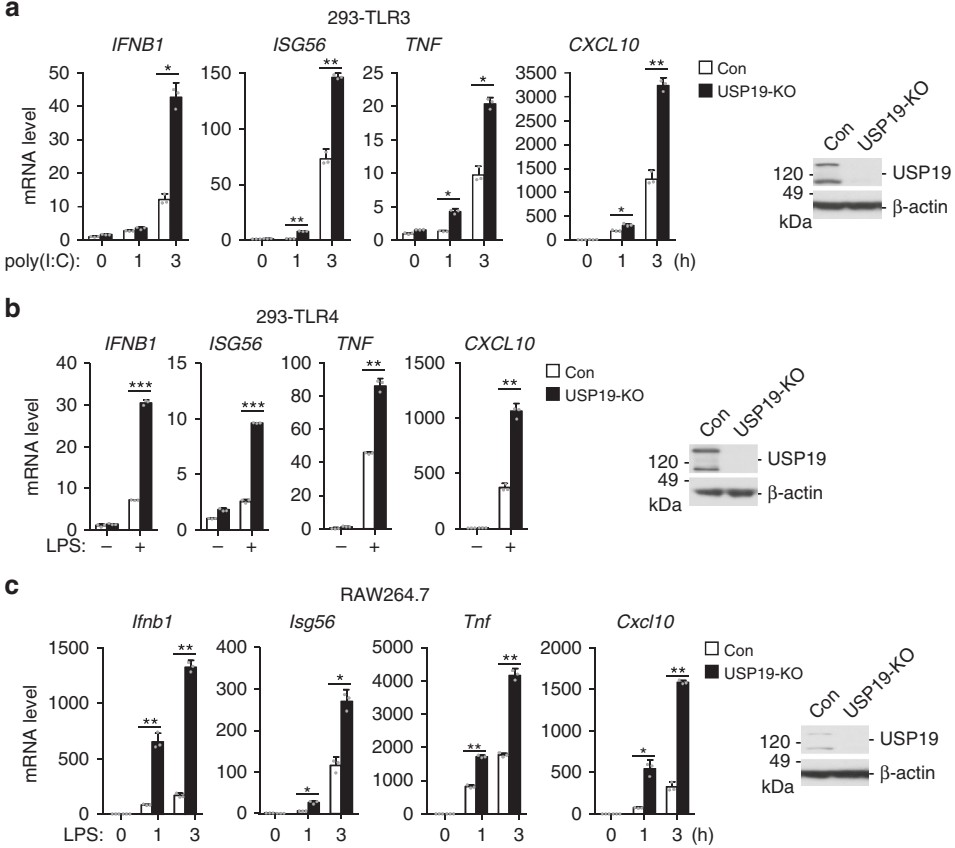

**Fig. 1** USP19 negatively regulates TLR3/4-mediated signaling. **a** Effects of USP19-deficiency on transcription of downstream genes induced by poly(I:C). USP19-deficient (USP19-KO) 293-TLR3 cells were generated by the CRISPR-Cas9 method. USP19-KO and control cells were treated with poly(I:C) (50 μg/ml) or left untreated for the indicated times. The mRNA levels of the indicated genes in the cells were measured by qPCR experiments. USP19 levels in the cells were analyzed by immunoblots (right blots). **b**, **c** Effects of USP19-deficiency on transcription of downstream genes induced by LPS. USP19-deficient (USP19-KO) 293-TLR4 or RAW264.7 cells were generated by the CRISPR-Cas9 method. The indicated USP19-KO and control cells were treated with LPS (100 ng/ml) or left untreated for 3 h. The mRNA levels of the indicated genes in the cells were measured by qPCR experiments. USP19 levels in the cells were analyzed by immunoblots (right blots). Graphs show mean ± SD; $n = 3$ independent samples. *$P < 0.05$, **$P < 0.01$, ***$P < 0.001$ (unpaired $t$-test). Data are representative of three experiments with similar results. Source data are provided as a Source Data file. Error bars represent standard deviation of the mean

D-galactosamine or LPS. Since poly(I:C) alone is insufficient to cause inflammatory death of mice, D-galactosamine is an agent used to enlarge systemic inflammatory responses of mice injected with poly(I:C)[17,18]. As shown in Fig. 3a, poly(I:C)- and LPS-induced production of IFN-β, TNF, IL-6, and CXCL10 was significantly increased in the sera of $Usp19^{-/-}$ compared to $Usp19^{+/+}$ mice. Hematoxylin-eosin (HE) staining analysis indicated that more serious inflammation was observed in the lungs of $Usp19^{-/-}$ mice after injected with poly(I:C) plus D-galactosamine or LPS (Fig. 3b). Comparing to the $USP19^{+/+}$ littermates, $Usp19^{-/-}$ mice were more susceptible to inflammatory death induced by poly(I:C) plus D-galactosamine or LPS (Fig. 3c). After administration of *Salmonella typhimurium*, the livers and spleens of $Usp19^{-/-}$ mice carried less bacteria than that of the $Usp19^{+/+}$ littermates. These results indicated that $Usp19^{-/-}$ mice had increased ability to clear *Salmonella typhimurium* comparing to their wild-type littermates (Fig. 3d). In addition, $Usp19^{-/-}$ mice produced higher levels of serum TNF and IL-6 (Fig. 3e), had intensified inflammation in the small intestinal villus (Fig. 3f) and more dramatic loss of body weights (Fig. 3g), and was more susceptible to inflammatory death after administration of *Salmonella typhimurium* (Fig. 3h). These data suggest that USP19 attenuates TLR3/4-mediated inflammatory responses in mice.

**USP19 removes K27-linked polyubiquitin moieties from TRIF.** We next determined the molecular mechanisms responsible for the inhibitory effects of USP19 on TLR3/4-mediated inflammatory responses. We firstly examined the effects of USP19 on signaling mediated by the shared components of TLR3/4-triggered pathways. Reporter assays indicated that USP19 inhibited ISRE activation mediated by overexpression of TRIF but not TBK1 or IRF3 (Fig. 4a). USP19 interacted with TRIF but not TLR3, TLR4, TRAM, MyD88, or TBK1 in mammalian overexpression systems (Fig. 4b). Domain mapping experiments indicated that the TIR (aa348–580) and the C-terminal domain (aa532–712) of TRIF could independently interact with USP19. On the other hand, the C-terminal USP but not the N-terminal tandem CHORD-SGT1 (CS) domains of USP19 had the ability to interact with TRIF (Fig. 4c). Consistently, USP19 mutants lack of its USP domain also lost their abilities to inhibit poly(I:C)- and LPS-triggered IFN-β promoter activation (Fig. 4d). Endogenous USP19 was associated with TRIF in unstimulated cells and their association was increased after poly(I:C) or LPS stimulation (Fig. 4e). These results suggest that USP19 acts at the level of TRIF.

Since USP19 is a deubiquitinating enzyme, we next determined whether it deubiquitinates TRIF. We found that overexpression of USP19 but not its enzymatic inactive mutant USP19(C607S)

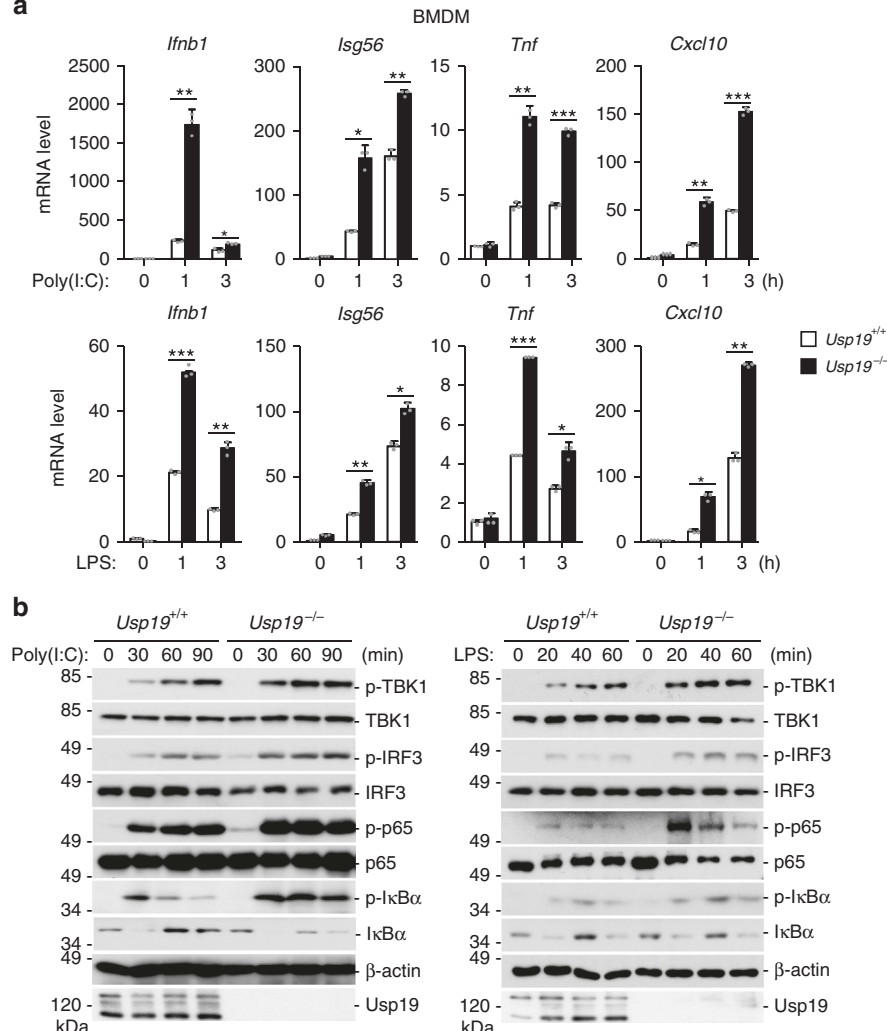

**Fig. 2** USP19 negatively regulates TLR3/4-mediated signaling in primary cells. **a** Effects of USP19-deficiency on poly(I:C)- and LPS-induced transcription of downstream genes. $Usp19^{+/+}$ and $Usp19^{-/-}$ BMDMs were stimulated with poly(I:C) (50 μg/ml), LPS (50 ng/ml) for the indicated times before qPCR with the indicated gene primers was performed. Graphs show mean ± SD; $n = 3$ independent samples. *$P < 0.05$, **$P < 0.01$, ***$P < 0.001$ (unpaired $t$-test). **b** Effects of USP19-deficiency on poly(I:C)- and LPS-induced phosphorylation of TBK1, p65, IRF3, and IκBα. $Usp19^{+/+}$ and $Usp19^{-/-}$ BMDMs were stimulated with poly(I:C) (100 μg/ml) or LPS (100 ng/ml) for the indicated times before immunoblots were performed with the indicated antibodies. Data are representative of three experiments with similar results. Source data are provided as a Source Data file. Error bars represent standard deviation of the mean

removed the polyubiquitin moieties from TRIF (Fig. 5a). Conversely, polyubiquitination of TRIF was increased in $Usp19^{-/-}$ compared to $Usp19^{+/+}$ BMDMs following poly(I:C) or LPS stimulation (Fig. 5b). Consistently, USP19 but not USP19 (C607S) inhibited poly(I:C)- and LPS-induced activation of the IFN-β promoter in reporter assays (Supplementary Fig. 4a). Reconstitution of USP19 but not USP19 (C607S) into $Usp19^{-/-}$ MLFs inhibited poly(I:C)- and LPS-induced transcription of $Ifnb1$ and $Cxcl10$ (Supplementary Fig. 4b). These data suggest that USP19 functions by deubiquitinating TRIF.

We next determined the types of ubiquitin chains of TRIF that were removed by USP19. By co-transfection of TRIF with ubiquitin mutants that contains only a single lysine residue (KO), we found that USP19 removed K27-linked polyubiquitin moieties from TRIF (Fig. 5c). By co-transfection of TRIF with ubiquitin mutants in which only one lysine reside is mutated to arginine, we found that USP19 failed to remove polyubiquitin moieties from TRIF only when K27 was mutated to arginine (Fig. 5c). These results suggest that USP19 deubiquitinates K27-linked polyubiquitin moieties from TRIF. Consistently, K27-linked

polyubiquitination of TRIF was profoundly increased in $USP19^{-/-}$ in comparison to $USP19^{+/+}$ cells following poly(I:C) or LPS stimulation (Fig. 5d). These results suggest that USP19 specifically removes K27-linked polyubiquitin moieties from TRIF following TLR3/4 activation.

**Deubiquitination of TRIF impairs its recruitment to TLR3/4.** We next determined how USP19-mediated deubiquitination of TRIF impairs TLR3/4 signaling. In mammalian overexpression systems, USP19 disrupted the interaction between TRIF and TLR3, but had no marked effects on the interaction between TRIF and TBK1 or TRAF3 (Fig. 6a). In similar experiments, USP19 (C607S) had no marked effects on both TRIF-TLR3 and TRIF-TBK1 interactions (Fig. 6b). Endogenous coimmunoprecipitation experiments indicated that more TRIF was recruited to TLR3 in USP19-deficient 293-TLR3 cells following poly(I:C) stimulation (Fig. 6c). These results suggest that USP19 impairs the association of TRIF with TLR3 after TLR3 activation in a deubiquitinating enzymatic activity dependent manner.

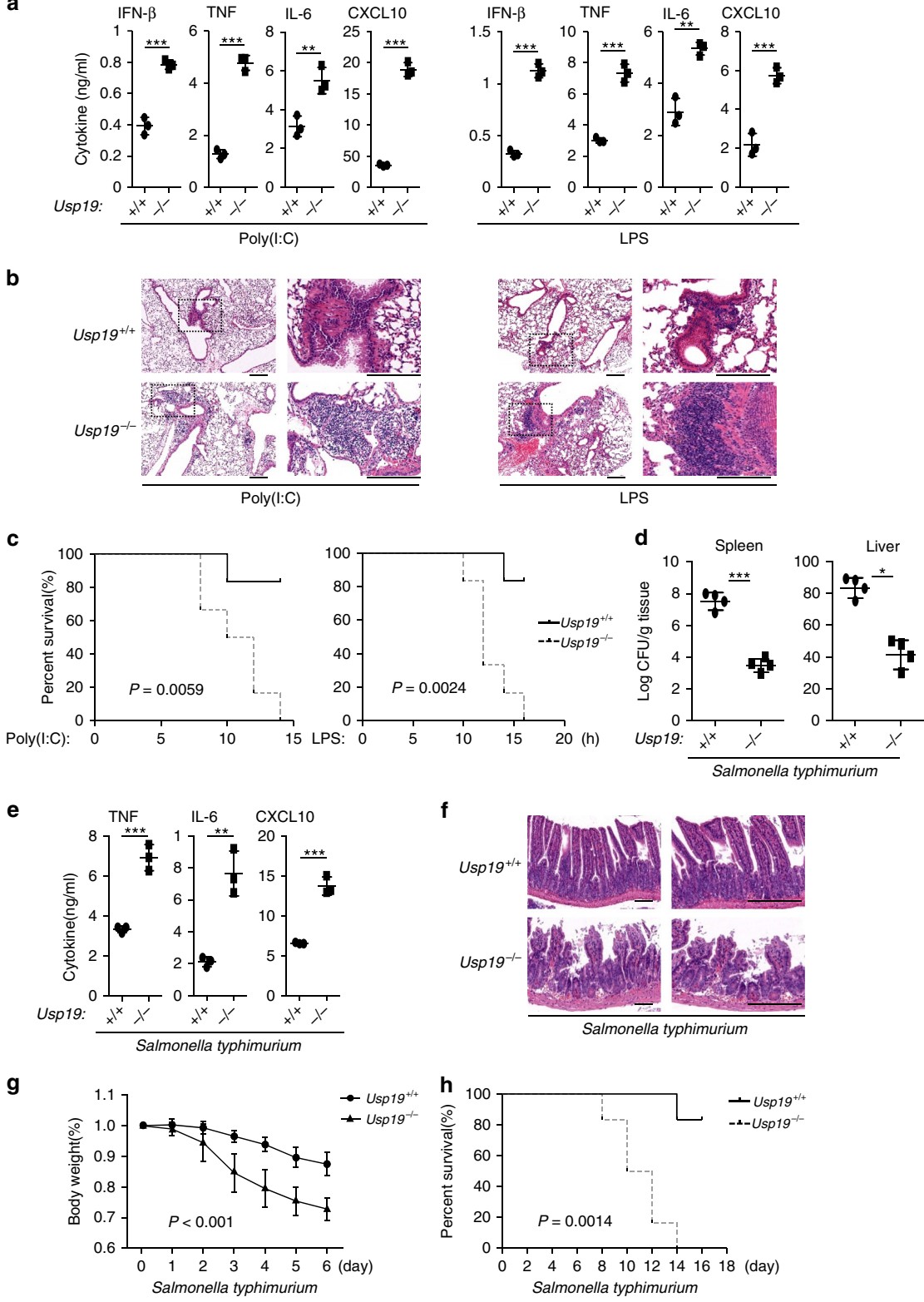

Similarly, we found that overexpression of USP19 but not USP19(C607S) inhibited the association between TRIF and TLR4 (Fig. 6d). In these experiments, USP19 had no marked effects on the association between TLR4 and TRIF-related adapter molecule (TRAM) (Fig. 6d), which acts as a link between TLR4 and TRIF[19]. As expected, USP19 but not USP19(C607S) inhibited the interaction between TRAM and TRIF (Fig. 6d). Consistently, TRIF–TLR4 and TRIF–TRAM associations were markedly

increased in USP19-deficient cells after LPS stimulation (Fig. 6e). Taken together, these results suggest that USP19 inhibits the recruitment of TRIF to TLR3/4 by removing K27-linked polyubiquitin moieties from TRIF.

**Cullin-3–Rbx1–KCTD10 catalyzes TRIF K27 polyubiquitination.** Since USP19-mediated removal of K27-linked polyubiquitin moieties from TRIF negatively regulates TLR3/4-triggered innate

**Fig. 3** USP19 negatively regulates TLR3/4-mediated response in vivo. **a** Sex- and age-matched mice were injected intraperitoneally with poly(I:C) (2 µg/g) plus ᴅ-galactosamine (1 mg/g) or LPS (10 µg/g) for 3 h before the indicated cytokine concentrations in the sera were determined by ELISA. Results are represented as mean ± SD, $n = 3$ per strain. **$P < 0.01$, ***$P < 0.001$ (unpaired $t$-test). **b** The mice were injected with poly(I:C) plus ᴅ-galactosamine or LPS as in **a** for 6 h before lung sections were used for histological analysis (H&E staining). Scale bars, 200 µm. **c** The mice were treated with poly(I:C) plus ᴅ-galactosamine or LPS as in **a**. The survival rates of the mice were recorded every 2 h in the following 20 h. Results are represented as mean ± SD, $n = 6$ per strain. $P$ values were from log-rank test. **d** The mice were orally infected with *Salmonella typhimurium* ($1 \times 10^7$ pfu per mouse), and bacterial loads were assessed in the spleen and liver 8 days post infection. Results are represented as mean ± SD, $n = 4$ per strain. *$P < 0.05$, ***$P < 0.001$ (unpaired $t$-test). **e** The mice were orally infected with *Salmonella typhimurium* as in **d** for 6 days, followed by measurement of the levels of inflammatory cytokines in the sera. Results are represented as mean ± SD, $n = 3$ per strain. **$P < 0.01$, ***$P < 0.001$ (unpaired $t$-test). **f** The mice were orally infected with *Salmonella typhimurium*, and the intestines of mice were used for histological analysis 6 days after infection (H&E staining). Scale bars, 300 µm. **g** The mice were orally infected with *Salmonella typhimurium*, and their body weights were monitored every day for 7 days. Results are shown as mean ± SD, $n = 6$ per strain. $P < 0.001$ (one-way ANOVA followed by Tukey's test). **h** The mice were orally infected with *Salmonella typhimurium*, and their survival rates were monitored every day for 18 days. Results are shown as mean ± SD, $n = 6$ per strain. $P$ value = 0.0014 (log-rank test). Data are representative of two experiments with similar results. Source data are provided as a Source Data file. Error bars represent standard deviation of the mean

immune and inflammatory responses, we sought to identify the E3 ubiquitin ligase(s) that mediate K27-linked polyubiquitination of TRIF. We screened a cDNA library containing 352 individual expression clones of ubiquitin-related proteins by reporter assays. These efforts led to the identification of eight candidates which have the abilities to potentiate poly(I:C)-triggered and TRIF-mediated activation of the IFN-β promoter (Supplementary Fig. 5a, b). We then used CRISPER/Cas9 technology to knockout these candidate proteins in 293-TLR3 cells. We found that deficiency of KCTD10, HERC4, RNF13, or RNF72 but not other tested ubiquitin-related proteins markedly inhibited poly(I:C)-triggered transcription of *IFNB1* gene (Fig. 7a). In mammalian overexpression systems, KCTD10 strongly mediated K27-linked polyubiquitination of TRIF, while HERC4, RNF13, and RNF72 had minimal effects (Fig. 7b). Endogenous coimmunoprecipitation experiments indicated that KCTD10 was associated with TRIF in un-stimulated cells, and the association was increased at the early time points (10 or 15 min) and then decreased to basal levels at the later time points (after 20 or 30 min) after poly(I:C) or LPS stimulation (Fig. 7c, d). Interestingly, the association of TRIF with USP19 was increased at the later time points after poly (I:C) or LPS stimulation (Fig. 7c, also see Fig. 4e). In addition, poly(I:C)- and LPS-induced K27-linked polyubiquitination of TRIF was markedly decreased in KCTD10-deficient cells compared with control cells (Fig. 7e). KCTD10 is a substrate-specific adapter in the Cullin-3-Rbx1-KCTD10 RING E3 ligase complex[20,21]. Consistently, KCTD10-mediated K27-linked polyubiquitination of TRIF was abolished in Cullin3- and Rbx1-deficient cells, suggesting that KCTD10-mediated polyubiquitination of TRIF is dependent on Cullin-3 and Rbx1 (Fig. 7f). qPCR experiments indicated that poly(I:C)-induced transcription of *IFNB1* and *CXCL10* genes was inhibited in $KCTD10^{-/-}$, $Cullin-3^{-/-}$, and $Rbx1^{-/-}$ in comparison to control cells (Supplementary Fig. 5c). Deficiency of KCTD10 also inhibited LPS- but not PGN- or R848-induced transcription of downstream genes (Supplementary Fig. 5d). Consistently, deficiency of KCTD10 inhibited poly(I:C)- and LPS-induced phosphorylation of TBK1 and p65 (Supplementary Fig. 5e). These results suggest that the Cullin-3-Rbx1-KCTD10 E3 ubiquitin ligase complex mediates TLR3/4-mediated signaling by catalyzing K27-linked polyubiquitination of TRIF.

**Lys523 of TRIF is modified by K27-linked polyubiquitination.** We next investigated which lysine residues in TRIF are targeted by Cullin-3-Rbx1-KCTD10 E3 ligase complex for K27-linked polyubiquitination. We carried out a systematic lysine (K) to arginine (R) mutation scanning and tested the effects of USP19 on their activation of ISRE in reporter assays. As shown in Fig. 8a, USP19 markedly inhibited ISRE activation mediated by wild-type

and all of the tested mutants of TRIF, with the exception of TRIF (K523R). In these experiments, we also noticed that TRIF(K523R) had markedly reduced ability in activating ISRE in comparison to wild-type and other examined mutants (Fig. 8a). We further investigated the functions of the TRIF(K523R) by reconstituting it into TRIF-deficient 293-TLR3 cells. We found that TRIF(K523R) partly lost the ability to mediate poly(I:C)-triggered induction of downstream *IFNB1* and *CXCL10* genes compared to wild-type TRIF and TRIF(K529R) (Fig. 8b). Further experiments indicated that mutation of K523 but not other examined lysine residues of TRIF to arginine abolished its K27-linked polyubiquitination (Fig. 8c). In similar experiments, K63-linked polyubiquitination of TRIF was comparable between wild-type TRIF and its K523R, K415R and K529R mutants (Fig. 8d). Moreover, KCTD10 catalyzed K27-linked polyubiquitination of wild-type TRIF and TRIF (K529R) but not TRIF(K523R) (Fig. 8e). Taken together, these results suggest that the Cullin-3-Rbx1-KCTD10 E3 ligase catalyzes K27-linked polyubiquitination of TRIF at K523.

## Discussion

TRIF is an essential adapter protein for TLR3/4-mediated innate immune responses. The activity and availability of TRIF are strictly controlled by several posttranslational modifications to exert sufficient protective immunity and avoid excessive immune damage after certain viral and bacterial infection. Although several E3 ubiquitin ligases have been reported to regulate TRIF polyubiquitination, it remains unknown on how deubiquitination of TRIF is regulated and whether such a regulation modulates TRIF-mediated innate immune responses. In this study, we have identified the first deubiquitinating enzyme, USP19, which mediates deubiquitination of TRIF and regulates TRIF-mediated innate immune responses.

Several lines of evidence support a key role of USP19 in TRIF-mediated signaling. Overexpression of USP19 inhibited poly(I:C)- and LPS-induced activation of NF-κB and IRF3, whereas USP19-deficiency had the opposite effects. Deficiency of USP19 increased serum cytokine levels after administration of poly(I:C) and LPS, and promoted inflammatory death caused by administration of poly(I:C) or LPS, or infection of *Salmonella typhimurium*. Previously, it has been shown that USP19 negatively regulates RNA virus-triggered type I IFN induction by deubiquitinating Beclin-1 and impairing RIG-I-MAVS/VISA association in certain human cell lines[22]. However, our results indicated that USP19-deficiency did not affect induction of downstream effector genes after infection with both RNA and DNA viruses or stimulation with TLR2 and TLR7/8 ligands in mouse primary immune cells. It is possible that human and mouse USP19 has distinct functions.

Coimmunoprecipitation experiments indicated that USP19 was associated with TRIF, and this association was increased upon

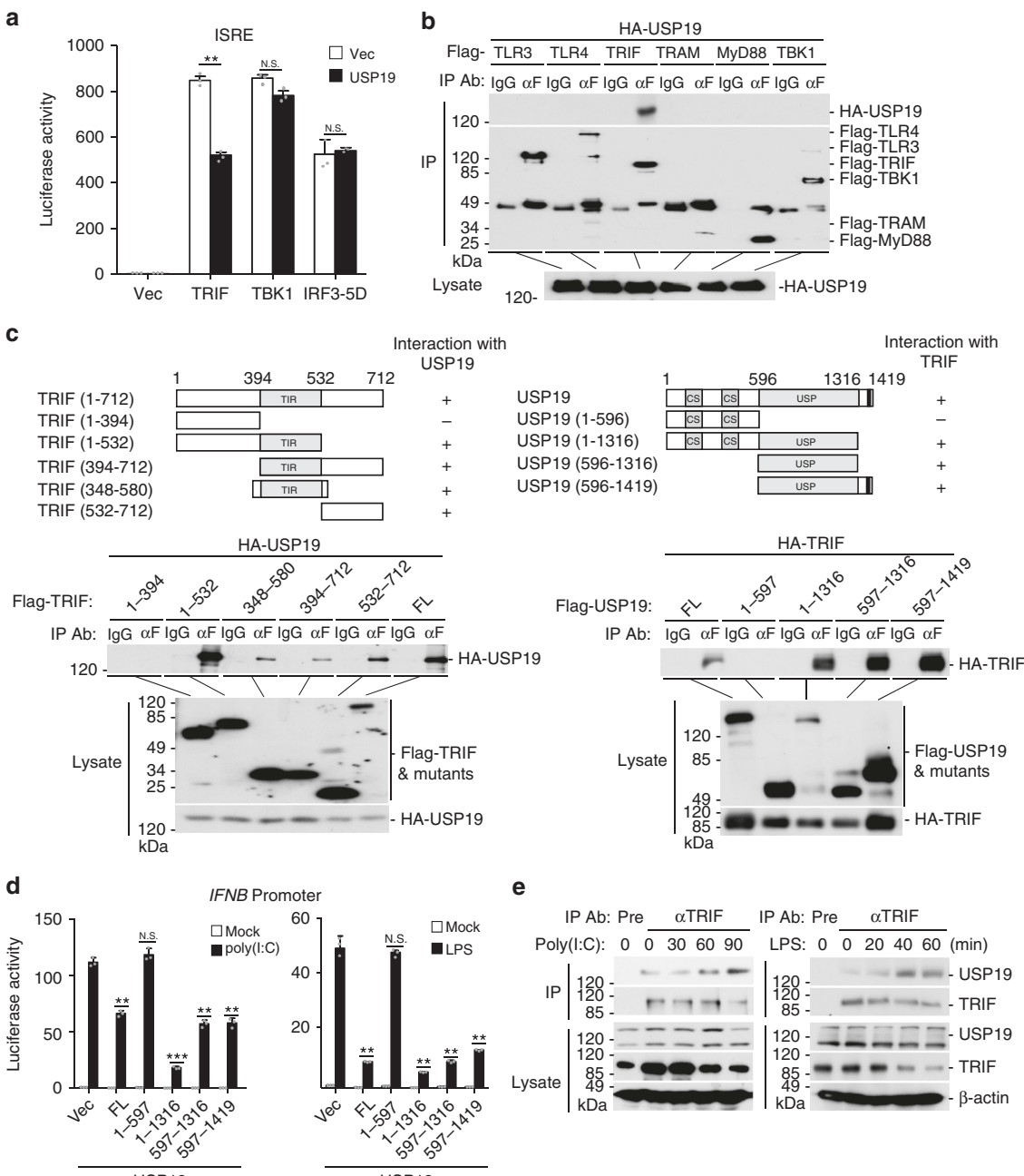

**Fig. 4** USP19 acts at the level of TRIF. **a** Overexpression of USP19 inhibits TRIF-mediated ISRE activation. HEK293 cells were transfected with ISRE reporter and the indicated adapter expression plasmids together with a control or USP19 expression plasmid for 24 h before luciferase assays were performed. Graphs show mean ± SD; $n = 3$ independent samples. **$P < 0.01$ (unpaired $t$-test); N.S., not significant. **b** USP19 interacts with TRIF in overexpression systems. HEK293 cells were transfected with HA-USP19 and the indicated adapter expression plasmids. Twenty hours after transfection, co-immunoprecipitation was performed with anti-Flag or control IgG. The immunoprecipitates and lysates were analyzed by immunoblotting with anti-HA or anti-Flag. **c** Domain mapping of the interaction between USP19 and TRIF. HEK293 cells were transfected with the indicated truncations before co-immunoprecipitation and immunoblotting analysis with the indicated antibodies. The schematic presentations of USP19 and TRIF truncations are shown at the top. **d** Effects of USP19 and its truncations on poly(I:C)- and LPS-induced IFN-β promoter activation. The 293-TLR3 or 293-TLR4 cells were transfected with the IFN-β promoter reporter and Flag-USP19 or its truncations for 24 h, and then left untreated, treated with poly(I:C) (50 μg/mL) or LPS (100 ng/ml) respectively for 8 h before luciferase assays. Graphs show mean ± SD; $n = 3$ independent samples. **$P < 0.01$, ***$P < 0.001$ (one-way ANOVA followed by Dunnett's test); N.S., not significant. **e** Endogenous USP19 is associated with TRIF. The 293-TLR3 or 293-TLR4 cells were left untreated or treated with poly (I:C) (100 μg/ml) or LPS (100 ng/ml) respectively for the indicated times. Cell lysates were immunoprecipitated with anti-TRIF or control IgG. The immunoprecipitates and lysates were analyzed by immunoblotting with indicated antibodies. Data are representative of three experiments with similar results. Source data are provided as a Source Data file. Error bars represent standard deviation of the mean

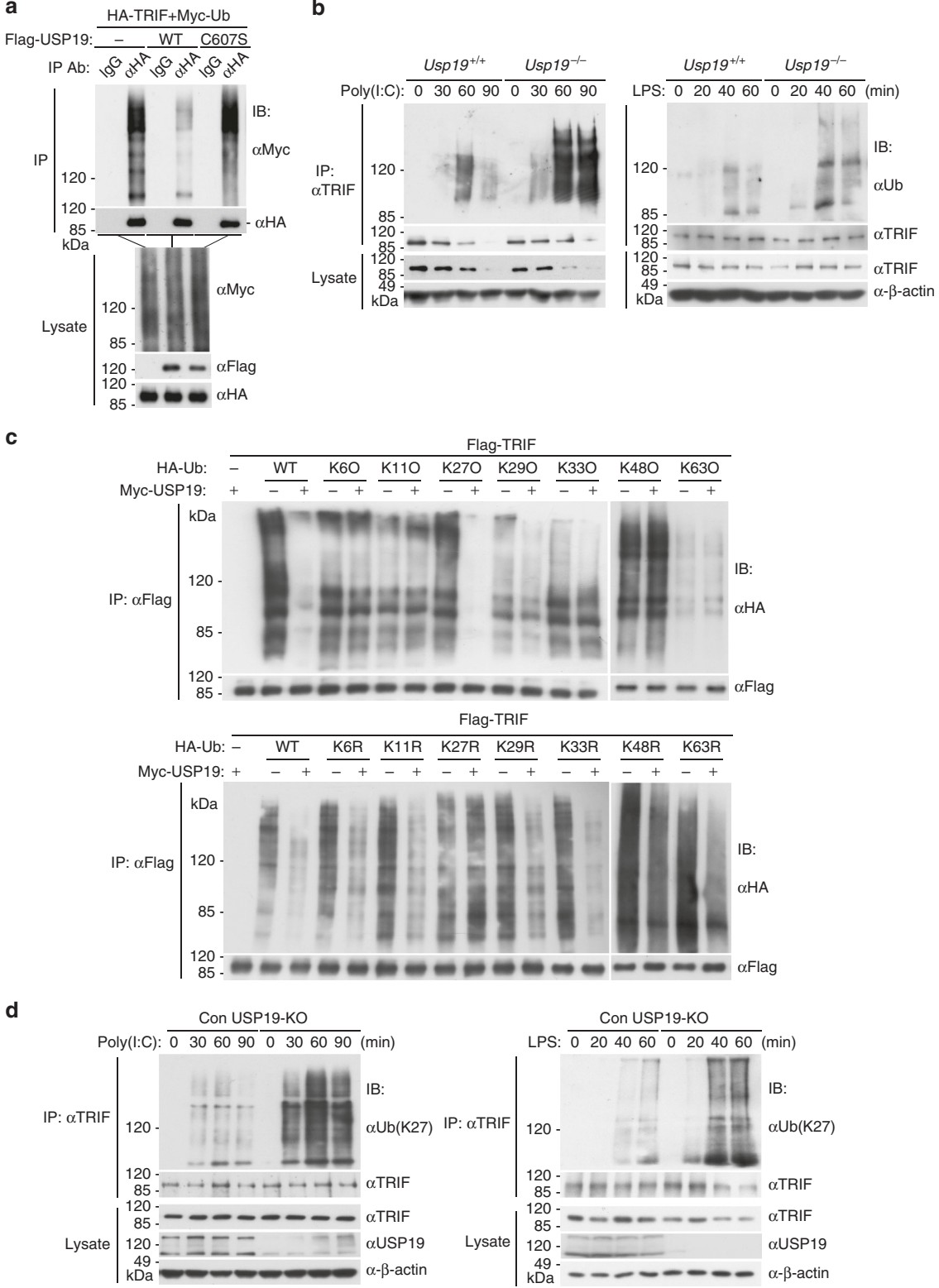

stimulation with poly(I:C) and LPS. Overexpression of USP19 but not its inactive mutant USP19(C607S) removed K27-linked but not other types of polyubiquitin moieties from TRIF, whereas USP19-deficiency increased poly(I:C)- and LPS-induced K27-linked polyubiquitination of TRIF. Further experiments indicated that wild-type USP19 but not USP19(C607S) impaired the interaction of TRIF with TLR3 or TLR4/TRAM, whereas USP19-deficiency increased the interactions. Our results suggest that

TRIF is modified by K27-linked polyubiquitination, and USP19 specifically removes K27-linked polyubiquitin moieties from TRIF following TLR3/4 activation, resulting in impairment of recruitment of TRIF to TLR3/4 and inhibition of innate immune responses.

Previously, it has been shown that USP19 is involved in many cellular processes including autophagy and immune response[23], endoplasmic reticulum-associated degradation[24], misfolding

**Fig. 5** USP19 removes K27-linked polyubiquitin moieties from TRIF. **a** USP19 but not its C607S mutant deubiquitinates TRIF. HEK293 cells were transfected with HA-TRIF and Myc-Ub together with empty vector, Flag-USP19 or Flag-USP19(CS). Twenty hours after transfection, ubiquitination assays were performed with the indicated antibodies. The detailed information of ubiquitination assays is described in the Methods. **b** USP19-deficiency potentiates polyubiquitination of TRIF after poly(I:C) and LPS stimulation. $Usp19^{+/+}$ and $Usp19^{-/-}$ BMDMs were left untreated or treated with poly(I:C) (100 μg/ml) or LPS (100 ng/ml) respectively for the indicated times. Cell lysates were immunoprecipitated with anti-TRIF. The immunoprecipitates and lysates were analyzed by immunoblotting with the indicated antibodies. **c** Overexpression of USP19 removes K27-linked polyubiquitin moieties from TRIF. HEK293 cells were transfected with Flag-TRIF, Myc-USP19, HA-ubiquitin or its mutants (KO, K-only; KR, K is mutated to R) together with a control and USP19 expression plasmid for 24 h before ubiquitination assays with the indicated antibodies. **d** USP19-deficiency potentiates poly(I:C)- and LPS-induced K27-linked polyubiquitination of TRIF. The 293-TLR3 cells or 293-TLR4 cells were left untreated or treated with poly(I:C) (100 μg/ml) or LPS (100 ng/ml) respectively for the indicated times before ubiquitination assays with the indicated antibodies. Data are representative of two or three experiments with similar results. Source data are provided as a Source Data file

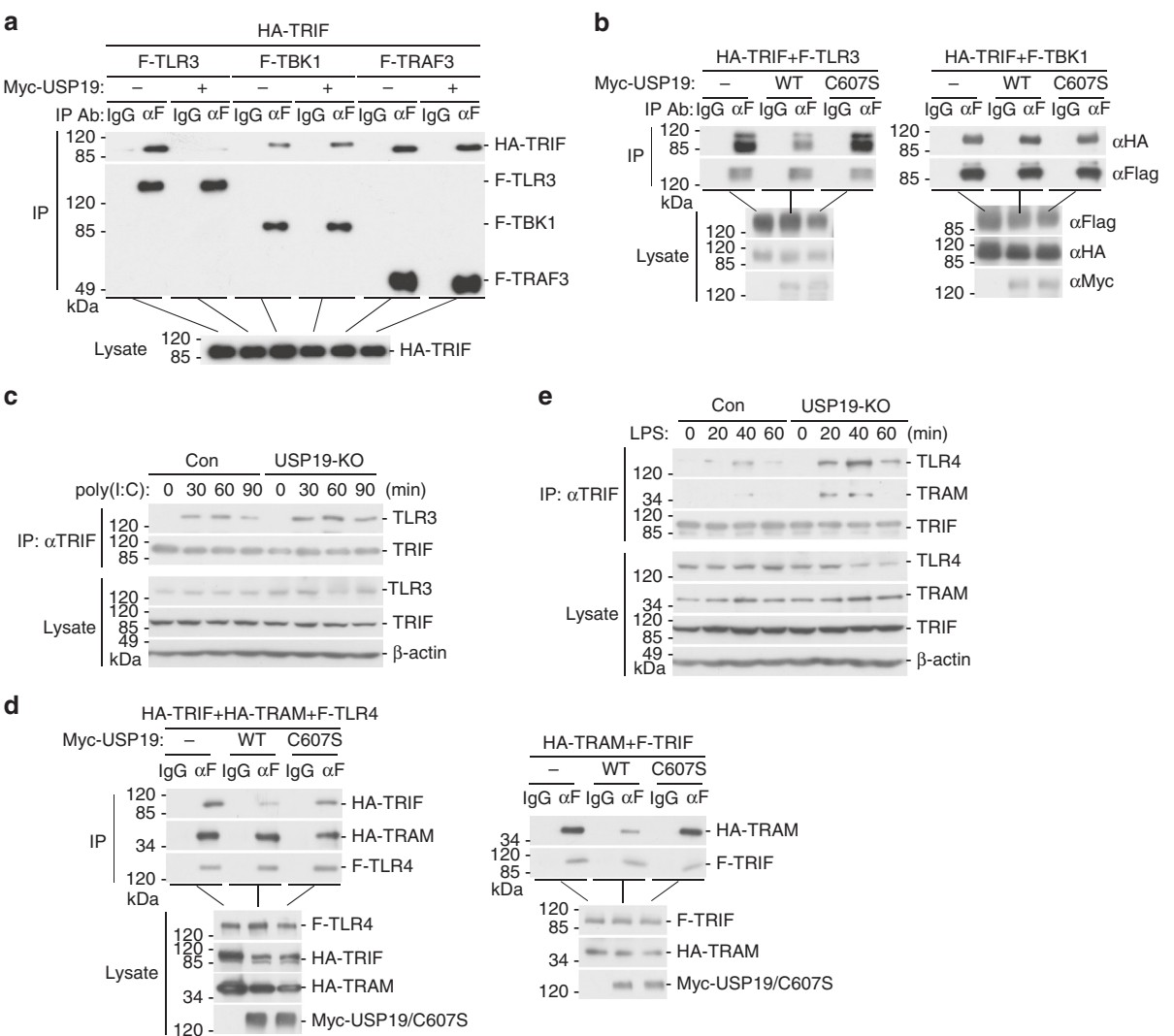

**Fig. 6** USP19-mediated deubiquitination of TRIF impairs its function. **a** Overexpression of USP19 inhibits the interaction of TRIF with TLR3 but not with TBK1 or TRAF3. HEK293 cells were transfected with Flag-TLR3, Flag-TBK1, or Flag-TRAF3 and HA-TRIF together with a control or Myc-USP19 expression plasmid for 20 h. Coimmunoprecipitation was performed with anti-Flag or control IgG. The immunoprecipitates and lysates were analyzed by immunoblotting with anti-HA or anti-Flag. **b** Overexpression of USP19 but not USP19(C607S) inhibits TRIF-TLR3 interaction. HEK293 cells were transfected with Flag-TLR3 or Flag-TBK1 and HA-TRIF together with a control or Myc-USP19 expression plasmid for 20 h before co-immunoprecipitation and immunoblotting analysis with the indicated antibodies. **c** USP19-deficiency potentiates TLR3-TRIF association. The control or USP19-deficient 293-TLR3 cells were treated with poly(I:C) (100 μg/ml) or left untreated for the indicated times. Cell lysates were immunoprecipitated with anti-TRIF or control IgG. The immunoprecipitates were analyzed by immunoblots with the indicated antibodies. **d** Overexpression of USP19 but not USP19(C607S) inhibits TRIF-TLR4 and TRIF-TRAM interactions. HEK293 cells were transfected with the indicated plasmids for 20 h before coimmunoprecipitation and immunoblotting analysis were performed with the indicated antibodies. **e** USP19-deficiency increases TLR4-TRIF association. The control or USP19-deficient 293-TLR4 cells were treated with LPS (100 ng/ml) or left untreated for the indicated times before coimmunoprecipitation and immunoblotting analysis with the indicated antibodies. Data are representative of three experiments with similar results. Source data are provided as a Source Data file

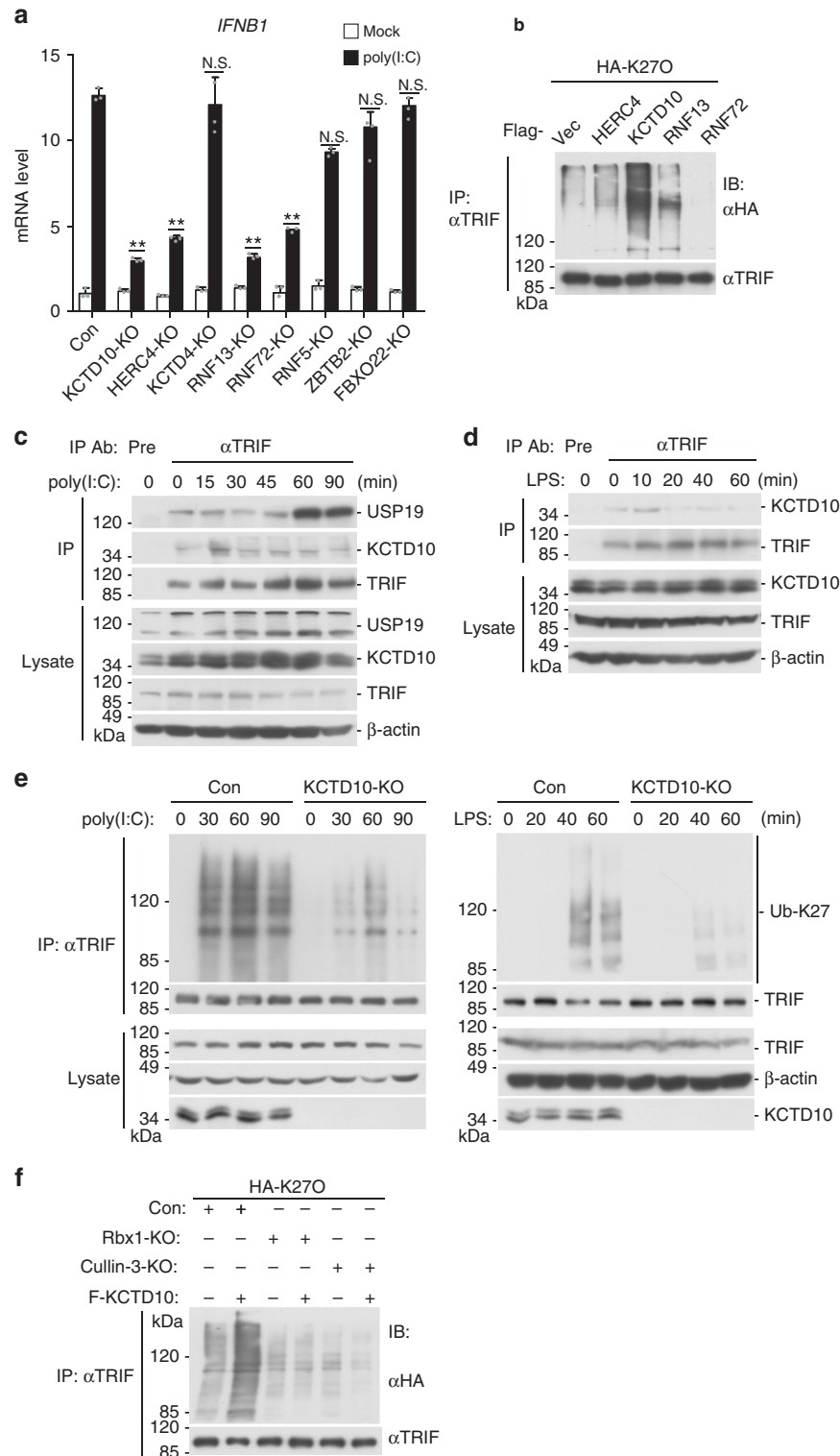

associated protein secretion[25], cell proliferation[26], hypoxia[27], muscle[28–32], and adipogenesis[33]. It has been reported that USP19 catalyzes the removal of various types of ubiquitin chains to regulate different physiological processes. For examples, USP19 negatively regulates type I IFN signaling pathway by blocking RIG-I-VISA interaction in a Beclin-1-dependent manner, in which. USP19 stabilizes Beclin-1 by removing its K11-linked polyubiquitin chains at lysine 437. USP19 removes K63-linked polyubiquitin moieties from HDAC1/2, which is crucial for regulation of HDAC1/2 activity in DNA damage repair[34]. USP19

deconjugates K63- and K27-linked polyubiquitin chains from TAK1 after TNF or IL-1β stimulation, leading to impairment of TAK1 activity and inhibition of TNF- and IL-1β-triggered inflammatory response[35]. USP19 deconjugates K48-linked polyubiquitin chains from HRD1, which prevents proteasomal degradation of HRD1 and ER-associated protein degradation (ERAD)[36].

Using an expression screen approach, we found that the Cullin-3-Rbx1-KCTD10 E3 ubiquitin ligase complex was responsible for mediating K27-linked polyubiquitination of TRIF.

**Fig. 7** Cullin-3–Rbx1–KCTD10 catalyzes K27-linked polyubiquitination of TRIF. **a** Effects of knockout of various ubiquitin-related proteins on poly(I:C)-induced transcription of *IFNB1* gene. The indicated ubiquitin-related genes were knockout in 293-TLR3 cells by CRISPR-Cas9 method. The cells were treated with poly(I:C) (50 μg/ml) or left untreated for 3 h before *IFNB1* mRNA was measured by qPCR experiments. Graphs show mean ± SD; $n = 3$ independent samples. **P < 0.01 (one-way ANOVA followed by Dunnett's test); N.S., not significant. **b** KCTD10 enhances K27-linked polyubiquitination of TRIF. The 293-TLR3 cells were transfected with HA-ubiquitin (K27O) and the indicated ubiquitin-related proteins expression plasmids for 20 h. The cells were then treated with poly(I:C) for 90 min. Cell lysates were immunoprecipitated with anti-TRIF. The immunoprecipitates were analyzed by immunoblots with the indicated antibodies. **c**, **d** Endogenous of TRIF with USP19 and KCTD10. The 293-TLR3 or 293-TLR4 cells were left untreated or treated with poly(I:C) (100 μg/ml) or LPS (100 ng/ml) respectively for the indicated times. Cell lysates were immunoprecipitated with anti-TRIF or IgG. The immunoprecipitates were analyzed by immunoblots with the indicated antibodies. **e** KCTD10-deficiency impairs K27-linked polyubiquitination of TRIF after poly(I:C) stimulation. The control or KCTD10-deficient 293-TLR3 cells were treated with poly(I:C) (100 μg/ml) or left untreated for the indicated times before ubiquitination assays were performed with the indicated antibodies. **f** Effects of Cullin-3- or Rbx1-deficiency on KCTD10-mediated K27-linked polyubiquitination of TRIF. The control, Cullin-3- or Rbx1-deficient 293-TLR3 cells were transfected with HA-K27O and KCTD10-Flag plasmids or empty vectors for 20 h. The cells were then treated with poly(I:C) (100 μg/ml) for 90 min. Cell lysates were immunoprecipitated with anti-TRIF. The immunoprecipitates were analyzed by immunoblots with anti-HA or anti-TRIF. Data are representative of two or three experiments with similar results. Source data are provided as a Source Data file. Error bars represent standard deviation of the mean

Deficiency of this complex impaired K27-linked polyubiquitination of TRIF as well as induction of downstream effector genes after poly(I:C) and LPS stimulation. Coimmunoprecipitation experiments indicated that the association of TRIF with KCTD10 was increased at the early time point and then decreased thereafter, whereas its association with USP19 was increased at the late time points. These results suggest that increased K27-linked polyubiquitination of TRIF by the Cullin-3-Rbx1-KCTD10 complex at the early phase of TLR3/4 stimulation promotes innate immune responses, while USP19-mediated deubiquitination of TRIF at the late phase terminates TLR3/4-triggered innate immune responses. In light of these observations, it would be interestingly to determine how USP19 enzymatic activities are regulated to deconjugate distinct linkage-types of polyubiquitin moieties of various substrates in different signaling pathways.

Our experiments also identified K523 of TRIF as the target residue for K27-linked polyubiquitination by the Cullin-3-Rbx1-KCTD10 complex. Mutation of K523 of TRIF to arginine impaired its K27-linked polyubiquitination as well as its ability to mediate downstream signaling. In addition, KCTD10 failed to catalyze K27-linked polyubiquitination. Recently, structural studies suggest that R522 and K523 of TRIF are crucial for its direct interaction with TRAM. The wedge-shaped surface containing R522, K523, Q518, and I519 of TRIF binds to the concave surface of TRAM TIR domain by electrostatic interaction[37]. In light of this observation, we propose that the wedge-shaped surface containing K523 not only directly interacts with TRAM through electrostatic interaction but also serves as a platform for K27-linked polyubiquitination catalyzed by the Cullin-3-Rbx1-KCTD10 E3 ligase complex.

In conclusion, our studies suggest that dynamic K27-linked polyubiquitination and deubiquitination of TRIF by the specific enzymes after TLR3/4 stimulation represent an important regulatory mechanism of TLR3/4-mediated innate immune and inflammatory responses.

## Methods

**Mice**. We generated *Usp19*-deficient mice by standard CRISPR/Cas9-mediated gene editing strategy. Gene sequencing results showed that a thymidine was inserted into the third exon of *Usp19* gene, which caused a reading-frame shift and the early translational termination of Usp19 after aa 42[35]. Mice were maintained in the special pathogen free facility of College of Life Sciences at Wuhan University. Eight-to-ten-week old, age- and sex-matched mice were used in all the experiments. Animals were handled according to the Guidelines of the China Animal Welfare Legislation, as approved by the Committee on Ethics in the Care and Use of Laboratory Animals of College of Life Sciences, Wuhan University.

**Reagents and antibodies**. Mouse monoclonal antibodies against Flag (Origene, 1:2000, F3165), HA (Origene, 1:2000, H6908), β-actin (Sigma, 1:10,000, A2228), Myc (CST, 1:1000, 5605), p-IκBα (CST, 1:1000, 9246 L); rabbit antibodies against

p-IRF3 (CST, 1:500, 37829), USP19 (Abcam, 1:1000, ab93159), TRIF (Abcam, 1:1000, ab180689), p65 (Santa Cruz Biotechnology, 1:1000, 71675), p-p65(S536) (CST, 1:1000, 3033),TBK1 (Abcam, 1:1000, ab40676) and p-TBK1 (Abcam, 1:1000, ab109272), ubiquitin (Abcam, 1:500, ab134953), K27-linkage specific polyubiquitin (Abcam, 1:1000, 181537), TLR3 (CST, 1:500, 6961), TLR4 (R&D, 1:500, AF1478), TRAM (Abcam, 1:500, ab96106), KCTD10 (Proteintech, 1:1000, 27279–1-AP); poly(I:C) (Invivogen), LPS (Sigma), R848 (Invivogen), PGN (Invivogen), human IFN-γ (Peprotech), murine M-CSF (Peprotech), Trizol (Takara Bio), SYBR Green (BIO-RAD), dual-specific luciferase assay kit (Promega, E1980), polybrene (Millipore,TR-1003-G), type II collagenase (Worthington), DNase I (Sigma-Aldrich), and D-galactosamine hydrochloride (D-Gal) (Sigma); and ELISA kits for TNF (Biolegend), IL-6 (Biolegend), CXCL10 (Boster) and IFN-β (PBL) were purchased from the indicated companies. HEK293 cells were obtained from ATCC. SeV (Cantell strain) (Charles River Laboratories), HSV-1 (KOS strain) (China Center for Type Culture Collection, Wuhan, China) were obtained from the indicated companies.

**Constructs**. Mammalian expression plasmids for Flag-, Myc- or HA-tagged USP19, KCTD10 and the other ubiquitin-related proteins, TRIF and their mutants were constructed by standard molecular biology techniques. Other plasmids used in this study were previously described[38–40].

**CRISPR-Cas9 knockout**. Genome engineering was performed utilizing the CRISPR-Cas9 system[41,42]. Double-stranded oligonucleotides corresponding to the target sequences were cloned into the lenti-CRISPR-V2 vector and co-transfected with packaging plasmids into HEK293 cells. Two days after transfection, the viruses were harvested and used to infect the indicated cells. The infected cells were selected with puromycin (1 μg/ml) for at least 5 days. The following sequences were targeted for human USP19 cDNA: 5′- GCAGAAGGATCGAGCAAACC-3′, mouse USP19 cDNA: 5′- GAGTCCTGGCGGCCCCCTCCT-3′, human KCTD10 cDNA: 5′- GACTCAACTACCTTCGAGACG-3′, mouse KCTD10 cDNA: 5′- CACGTTC AGCTTCACGTACT-3′, human Rbx1 cDNA: 5′-GCGCCGCTGTTGGTGCCG CT-3′, human Cullin-3 cDNA: 5′- GACCACTGTTATTCTTACGC-3′.

**Preparation of BMDMs and BMDCs**. The bone marrow cells were isolated from tibia and femur. For preparation of BMDMs, the bone marrow cells were cultured in RPMI 1640 medium which contains 10% FBS and murine M-CSF (10 ng/mL) for 5 days. For preparation of BMDCs, the bone marrow cells were cultured in RPMI1640 medium which contains 10% FBS and GM-CSF conditional medium for 9 days.

**Preparation of MLFs**. The lungs of mice (4–6 week-old) were minced, which were then digested with type II collagenase (10 μg/ml) and DNase I (20 μg/ml) in calcium and magnesium free HBSS at 37 °C for 3 h with shaking. Cell suspensions were sequentially filtered through 100 μm and 40 μm cell strainers, which were then centrifuged at 1500 rpm for 4 min. The pelleted cells were then cultured in 100 mm dishes in DMEM/Ham's F-12 (1:1 v/v) medium containing 10% FBS, 15 mM HEPES, 2 mM L-glutamine, 50 U/ml penicillin, and 50 μg/ml streptomycin. One hour later, the adherent fibroblasts were washed with HBSS and then cultured in the above medium.

**Administration of poly(I:C) and LPS**. Age- and sex-matched *Usp19*[+/+] and *Usp19*[−/−] mice were grouped for the experiments. The mice were injected intraperitoneally with poly(I:C) (2 μg/g body weight) and D-galactosamine (1 mg/g body weight), or with LPS (10 μg/g body weight). The survival status of the injected mice was monitored at half an hour interval.

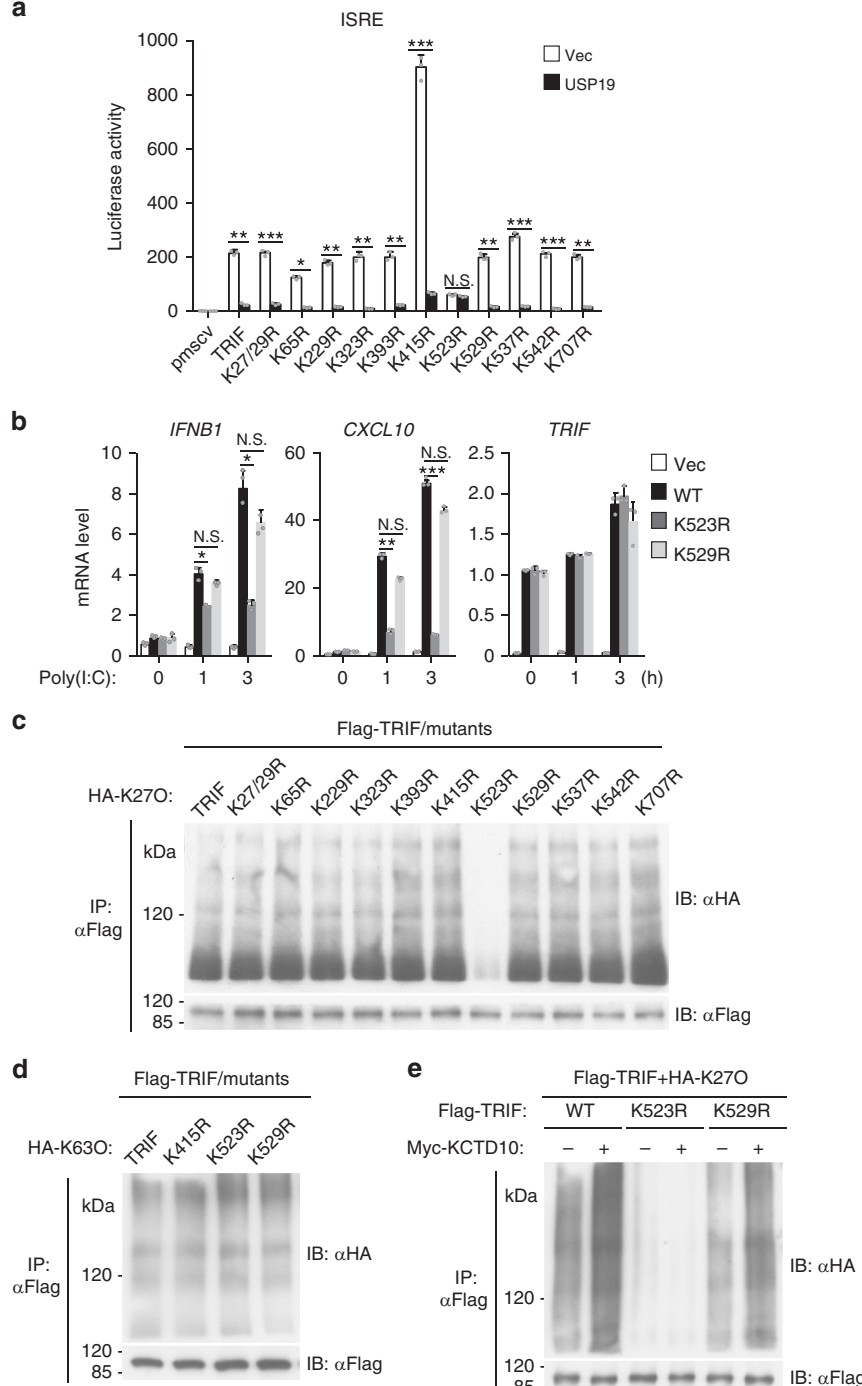

**Fig. 8** K523 of TRIF is modified by K27-linked polyubiquitination. **a** Effects of USP19 on ISRE activation induced by wild-type (WT) and mutant TRIF. HEK293 cells were transfected with ISRE reporter and WT or mutant TRIF expression plasmids together with a control or USP19 expression plasmid for 20 h before reporter assays were performed. Graphs show mean ± SD; *n* = 3 independent samples in **a**, **b**. *P < 0.05, **P < 0.01, ***P < 0.001 (one-way ANOVA followed by Dunnett's test). N.S., not significant. **b** Effects of TRIF and its mutants on poly(I:C)-induced transcription of downstream genes. TRIF-deficient 293-TLR3 cells reconstituted with TRIF or its mutants were treated with poly(I:C) or left untreated for the indicated times before qPCR analysis. Graphs show mean ± SD; n = 3 independent samples in **a**, **b**. *P < 0.05, **P < 0.01, ***P < 0.001 (one-way ANOVA followed by Turkey's test). N.S., not significant. **c** Effects of mutation of various lysine residues of TRIF on its K27-linked polyubiquitination. Flag-tagged TRIF or its mutants were individually transfected into HEK293 cells along with HA-ubiquitin (K27O). Twenty hours after transfection, ubiquitination assays were performed with the indicated antibodies. **d** Effects of mutation of various lysine residues of TRIF on its K63-linked polyubiquitination. Flag-tagged TRIF or its mutants were transfected into HEK293 cells together with HA-ubiquitin (K63O). The experiments were performed similarly as **c**. **e** Effects of KCTD10 on K27-linked polyubiquitination of WT and mutant TRIF. The experiments were performed similarly as **c**. Data are representative of two or three experiments with similar results. Source data are provided as a Source Data file. Error bars represent standard deviation of the mean

**Administration of *Salmonella typhimurium***. Age- and sex-matched *Usp19*[+/+] and *Usp19*[−/−] mice were grouped for the experiments. The mice were orally administrated with *Salmonella typhimurium* ($1 × 10^7$ pfu per mouse), then the body weights and survival status of the mice were examined every day.

**Serum cytokine levels**. The mice were administrated with poly(I:C) plus D-galactosamine, LPS or *Salmonella typhimurium* for the indicated times. The sera were then collected for measurements of levels of TNF, IL-6, CXCL10, and IFN-β by ELISA.

**Reporter assays**. HEK293 cells plated on 24-well dishes were transfected with the indicated reporter and other plasmids by calcium phosphate precipitation. For each transfection, pRL-TK *Renilla* luciferase reporter plasmid (0.01 μg) was added to serve as an internal control, and an empty plasmid was added to ensure that the same amount of plasmids was used for each transfection. The luciferase activities were measured with a dual-specific luciferase assay kit, and firefly luciferase activities were normalized with *Renilla* luciferase activities.

**Coimmunoprecipitation and ubiquitination experiments**. Cells were lysed in NP-40 lysis buffer (20 mM Tris-HCl pH 7.4, 150 mM NaCl, 1 mM EDTA, 1% NP-40, 10 μg/ml aprotinin, 10 μg/ml leupeptin, and 1 mM phenylmethylsulfonyl fluoride). The lysate was aliquoted (0.4-mL) for coimmunoprecipitation with the indicated antibody or control IgG (0.5 μg or 0.5 μl for antiserum) and Protein G Sepharose (25 μL of a 1:1 slurry) for 2 h. The beads were then washed for three times with lysis buffer containing 0.5 M NaCl. For ubiquitination experiments, the immunoprecipitates were re-extracted in lysis buffer containing 1% SDS and denatured by heating for 10 min. The samples were centrifuged at $12,000 × g$ for 1 min, and then the supernatants were diluted with lysis buffer until SDS concentration was decreased to 0.1%. The supernatants were then subjected to re-immunoprecipitation with the indicated antibodies. The immunoprecipitates were analyzed by immunoblots with the indicated ubiquitin antibodies.

**qPCR**. Total RNA was isolated and mRNA abundance of the indicated genes was measured by qPCR. Data shown are the relative abundance of the indicated mRNA normalized to that of GAPDH. Gene-specific primer sequences were listed in Supplementary Table 1.

**Immunohistochemistry**. Mouse lungs or intestines were excised, fixed in formalin and then embedded in paraffin. The embedded organs were sectioned (5 μm), which were placed on polylysine-coated slides. The sections were de-paraffinized in xylene, rehydrated through graded ethanol, quenched in 3% hydrogen peroxide, and heated by microwaving for 7 min in 10 mM citrate buffer (pH 6.0). The sections were then counterstained with hematoxylin for 5 min. Photos were taken using a HistoFAXS system.

**Statistical analysis**. Differences between experimental and control groups were determined by unpaired *t*-test (where two groups of data were compared) or by one-way ANOVA analysis (where more than two groups of data were compared). *P* values < 0.05 were considered statistically significant. For animal survival analysis, the Kaplan–Meier method was adopted to generate graphs, and the survival curves were analyzed with log-rank analysis.

**Reporting summary**. Further information on research design is available in the Nature Research Reporting Summary linked to this article.

## Data availability

All data supporting the findings of this study are available within the article and its supplementary information files, or may be obtained from the corresponding author upon reasonable request. The source data underlying Figs. 1a–c, 2a, 3a, 3d, 3e, 3g, 4a, 4d, 7a, 8a–b and Supplementary Figs. 1a–d, 2, 3a–b, 4a–b and 5a–d are provided in Supplementary Figs. 6–13 as a Source Data file. A reporting summary for this article is available as a Supplementary Information file.

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

## Acknowledgements

This work was supported by grants from the State Key R&D Program of China (2017YFA0505800, 2016YFA0502102), and the National Natural Science Foundation of China (31830024, 31630045, and 31870870). We thank Dr. Hong-Liang Li for providing 293-TLR4 cells, and Dr. Lin Guo for providing *Salmonella typhimurium*.

## Author contributions

X.W., C.L., and H.B.S. conceived and designed the study; X.W., C.L., T.X. and X.Z. performed the experiments; H.B.S., X.W., Q.Y. and C.L. analyzed the data. H.B.S., X.W., Q.Y. and C.L. wrote the manuscript.

## Additional information

**Competing interests:** The authors declare no competing interests.

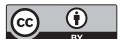

