## [Peer Review File · Nature Communications]

Reviewers' comments:

Reviewer #1 (Ubiquitination, inflammation)(Remarks to the Author):

The manuscript 'regulation of TRIF-mediated innate immune response by K27-linked polyubiquitination and deubiquitination', by Wu et al. nicely demonstrates the ability of USP19 to act as a K27-specific deubiquitinating enzyme (negatively) regulating TLR3/TLR4-TRIF mediated inflammatory signaling and production of inflammatory cytokines and type-1 interferons. The authors also identify the E3 ubiquitin ligase complex responsible for K27-linked polyubiquitination of TRIF.

The manuscript is concise, informative and clear, and data are overall convincing (some minor issues, see below), so I am definitely in favor of publication of this study in Nat. Communications.

Minor comments:

- Fig. 1a dose response: there is little difference in effect of the different doses of USP19 (especially for IFN- β and ISRE). Can you include a lower dose of USP19 to show loss of its inhibitory effect? This to have a real dose response.
- Fig. 1 e and f : why do you have a double band for USP19 on Western?
- Fig. 2b : for completeness, a Western blot for I κ B α should be included.
- Fig. 3b: why do you look at lung pathology when using D-Galactosamin, which is a hepatotoxin used to study liver failure?
- Fig. 5c : not really convincing. Based on this you could also claim USP19 specificity for K63 chains.
- Fig. 7c KCTD10 association with TRIF at early time points. This is not a convincing Western to make such claim. Replace by a better western.
- TNF α should be replaced by TNF (since TNF β has been renamed to LT).

Reviewer #2 (TLR signaling, inflammasome)(Remarks to the Author):

The authors identify USP19 as the enzyme that deubiquitinates TRIF after TLR3 or TLR4 activation, thereby increasing the production of inflammatory (and anti-inflammatory) cytokines. USP19 $^{-/-}$ mice show more cytokine production and more sensitivity to infection. The authors also claim that USP19 cleaves K27 ubiquitin linkages, and show that TRIF K523 is the site for K27 polyubiquitination by the Cullin-3-Rbx1-KCTD10 complex.

Overall, this work is solid. However, the experiments are overdocumented (much more should go in supplementary materials). Moreover, both introduction and discussion are minimally informative, and almost completely repetitive to what shown in results.

USP19 has been shown to modulate several inflammatory responses. It deubiquitinates a good deal of other signaling molecules, and is reported to cleave several sorts of polyubiquitin linkages. This information should be cited and discussed.

Specific suggestions:

- stats are wrong in that they extensively use t-tests in multiple comparisons, where one-way ANOVA is required instead.
- one cannot understand what the figures show by just reading the legends. Just one example: in Fig 1a the y axis reads Rel. Luc. Act, and there are 3 graphs labelled IFN- β , ISRE and NF- κ B. The legend says cells "were transfected with the indicated reporter", however IFN- β , ISRE and NF- κ B are not reporters but a cytokine, a DNA binding site and a transcription factor respectively. What was actually done? Such loose description is common throughout.

-western blots do not indicate molecular weights.

-why does the western blot of USP19 in transfected cells show two bands? Why no bands in non-transfected cells? USP19 is reportedly ubiquitous

Response to the reviewers' comments

Reviewer #1

General Comments

“The manuscript is concise, informative and clear, and data are overall convincing (some minor issues see below), so I am definitely in favor of publication of this study in Nat. Communications.”

Reply:

We thank the reviewer for liking our study and the very encouraging comments.

Minor Comments

1. Fig. 1a dose response: there is little difference in effect of the different doses of USP19 (especially for IFN- β and ISRE). Can you include a lower dose of USP19 to show loss of its inhibitory effect? This to have a real dose response.

Reply:

Following the reviewer's suggestion, we have now performed the experiments with lower doses of USP19. The new data has been included in the new Supplementary Fig. 1a and 1b and described in page 5. The new data shows a much better dose response as expected.

2. Fig. 1e and f : why do you have a double band for USP19 on Western ?

Reply:

USP19 has more than two dozens of splice variants/isoforms. We purchased the antibody against USP19 from Abcam (USP19, ab93159), which recognizes two bands as specified in the company's product sheet. As shown in the original figure 1e and f (now new Fig. 1a-c), this antibody did detect two bands in both human 293 and mouse Raw264.7 cells. However, both bands were not detected in USP19 knockout cells. The simplest explanation is that this antibody can detect two different splice variants of USP19, or alternatively, the lower band represents a cleaved fragment of USP19 isoform 1 (the upper band). We have described this in the text (page 6).

3. Fig. 2b : for completeness, a Western blot for I κ B should be included.

Reply:

Following the reviewer's suggestion, a Western blot for I κ B α has been included in the revised manuscript (see new Fig. 2b).

4. Fig. 3b: why do you look at lung pathology when using D-Galactosamine, which is a hepatotoxin used to study liver failure ?

Reply:

Although D-galactosamine is known to induce necrosis of the hepatocytes, two original studies by Galanos and coworkers^{1,2}, as well as several hundreds of publications, have shown that D-galactosamine can be used as a sensitizer for systemic inflammatory response³. Since poly(I:C) alone is insufficient to cause inflammatory death of mice, D-galactosamine is used to amplify systemic inflammatory response of mice injected with poly(I:C), which has also been used in various previous studies. The lung is a widely examined organ for systematic inflammatory response.

5. Fig. 5c :not really convincing. Based on this you could also claim USP19 specificity for K63 chains.

Reply:

We agree with the review that Fig. 5c alone could not lead to the conclusion. However, together with the other experiments shown in the paper, we could draw the conclusion that USP19 deubiquitinates K27-linked polyubiquitin moiety from TRIF. To make the conclusion more decisive, we performed additional experiments. By co-transfection of TRIF with ubiquitin mutants that contains only a single lysine residue (K-only, KO), we found that USP19 specifically removed K27-linked polyubiquitin moieties from TRIF (as shown in new Fig 5c, upper panels). Our original results showed that USP19 failed to remove polyubiquitin moieties from TRIF only when K27 was mutated to arginine (as shown in Fig 5c, lower panels). These results suggest that USP19 deubiquitinates K27-linked polyubiquitin moieties from TRIF.

6. Fig. 7c KCTD10 association with TRIF at early time points. This is not a convincing Western to make such claim. Replace by a better western.

Reply:

Following the reviewer's suggestion, we have now provided a better anti-KCTD10 immunoblot in Fig. 7c in the revised manuscript.

7. TNF α should be replaced by TNF (since TNF β has been renamed to LT).

Reply:

Following the review's suggestion, we have replaced "TNF α " with "TNF" in the revised manuscript.

Reviewer #2

“Overall, this work is solid. However, the experiments are overdocumented (much more should go in supplementary materials). Moreover, both introduction and discussion are minimally informative, and almost completely repetitive to what shown in results. USP19 has been shown to modulate several inflammatory responses. It deubiquitinates a good deal of other signaling molecules, and is reported to cleave several sorts of polyubiquitin linkages. This information should be cited and discussed.”

Reply:

We thank the reviewer for liking our study and good advices.

Following the reviewer’s suggestion, we have now moved some data to the Supplementary Information section, including the original Fig. 1a-c, Fig. 7g-i.

We have now revised the Introduction and Discussion sections. We have also cited and discussed USP19-related information in the revised manuscript (see page 15-16).

Specific points

1. Stats are wrong in that they extensively use t-tests in multiple comparisons, where one-way ANOVA is required instead.

Reply:

Following the reviewer’s suggestion, we have now corrected the stats of Fig. 3g and the original Fig. 1a-c (now new Supplementary Fig. 1a-c) from t-tests to one-way ANOVA in the revised manuscript. For the other experiments, even multiple groups of samples exist, our purpose is to compare control and each experimental group, and therefore, t-tests were used.

2. One cannot understand what the figures show by just reading the legends. Just one example: in Fig 1a the y axis reads Rel. Luc. Act, and there are 3 graphs labelled IFN-beta, ISRE and NF-kB. The legend says cells "were transfected with the indicated reporter", however IFN-beta, ISRE and NF-κB are not reporters but a cytokine, a DNA binding site and a transcription factor respectively. What was actually done? Such loose description is common throughout.

Reply:

We are sorry for having not described the experiments more clearly in the figure legends. We have now provided more precise description in the revised manuscript. IFN-β stands for IFN-β promoter reporter, ISRE and NF-κB stands for ISRE and NF-κB reporter respectively.

3. Western blots do not indicate molecular weights.

Reply:

We have now added molecular weights following the reviewer’s suggestion (the

raw data in the Supplementary Information section is also labelled with molecular weights).

4. *Why does the western blot of USP19 in transfected cells show two bands? Why no bands in non-transfected cells? USP19 is reportedly ubiquitous.*

Reply:

USP19 has more than two dozens of splice variants/isoforms. We purchased the antibody against USP19 from Abcam (USP19, ab93159), which recognizes two bands of endogenous proteins as specified in the company's product sheet. As shown in the original figure 1e and f (now new Fig. 1a-c), this antibody did detect two bands in both human 293 and mouse Raw264.7 cells. However, both bands were not detected in USP19 knockout cells. The simplest explanation is that this antibody can detect two different splice variants of USP19, or alternatively, the lower band represents a cleaved fragment of USP19 isoform 1 (the upper band). We have described this in the text (page 6).

In transfected cells, the USP19 isoform 1 was tagged with HA tag. The transfected USP19 was detected with anti-HA, therefore, only one band was detected. Because no HA-tagged USP19 was expressed in un-transfected cells, therefore, no bands were detected in non-transfected cells.

Reference

1. Galanos, C., Freudenberg, M.A. & Reutter, W. Galactosamine-induced sensitization to the lethal effects of endotoxin. *Proc Natl Acad Sci U S A* **76**, 5939-5943 (1979).
2. Lehmann, V., Freudenberg, M.A. & Galanos, C. Lethal toxicity of lipopolysaccharide and tumor necrosis factor in normal and D-galactosamine-treated mice. *The Journal of experimental medicine* **165**, 657-663 (1987).
3. Silverstein, R. D-galactosamine lethality model: scope and limitations. *Journal of endotoxin research* **10**, 147-162 (2004).

Reviewers' comments:

Reviewer #1 (Remarks to the Author):

The authors adequately addressed all of my concerns, so I agree with the publication of the paper.

Reviewer #2 (Remarks to the Author):

The authors have responded satisfactorily to most of my queries in this revised version.
The paper is much more streamlined without losing any relevant information.

However, I noticed that the authors have omitted one piece of relevant info. They do not describe their Usp19 knockout mice, nor they state whether these mice are available as a mouse line. They should add a supplementary method to describe how they got and identified the mice, and a supplementary figure or table with the sequence of mutation(s).
This is only a minor revision.

Additionally, they keep analyzing at least some of the results of experiment with multiple groups by t-tests, where one-way anova is required, followed possibly by post tests like Tukey's or Dunnett's. Using t tests is not correct, irrespective of what they believe.
Again this is just a minor revision, but should be done.

Point-by-point response

Reviewer #2:

“I noticed that the authors have omitted one piece of relevant info. They do not describe their Usp19 knockout mice, nor they state whether these mice are available as a mouse line. This is only a minor revision.”

Reply:

Generation and identification of *Usp19* gene knockout mice with a C57BL/6 background were performed by our laboratory as previous described ¹. We have now added the statement and related reference in the Method section (page 18).

-Additionally, they keep analyzing at least some of the results of experiment with multiple groups by t-tests, where one-way anova is required, followed possibly by post tests like Tukey's or Dunnett's. Using t tests is not correct, irrespective of what they believe.

Reply:

We have now corrected the stats in Fig 3g, Fig 8b, Supplementary Fig 1a-c and Supplementary Fig 4a-b using one-way ANOVA followed by Tukey's test in the revised manuscript. We have now corrected the stats of Fig 4d, Fig 7a, Fig 8a and Supplementary Fig 5a-c using one-way ANOVA followed by Dunnett's test in the revised manuscript. All changes are highlighted.

1. Lei, C.Q. & Wu, X. USP19 Inhibits TNF-alpha- and IL-1beta-Triggered NF-kappaB Activation by Deubiquitinating TAK1. **203**, 259-268 (2019).

REVIEWERS' COMMENTS:

Reviewer #2 (Remarks to the Author):

The authors have corrected all outstanding problems

Response to reviewers

The reviewers have no additional questions.